# More Reliable Land Price Index: Is There a Slope Effect?

Yi Huang [1,*] and Geoffrey Hewings [2]

1  Institute of Urban Development, Nanjing Audit University, Nanjing 211815, China
2  Regional Economics Applications Laboratory (REAL), Departments of Economics, Agricultural & Consumer Economics, Geography & Regional Science, Urban & Regional Planning and the Institute of Government & Public Affairs, University of Illinois at Urbana-Champaign, Champaign, IL 61820, USA; hewings@illinois.edu
*  Correspondence: yihuang@nau.edu.cn

**Abstract:** This paper focuses on the physical attributes of land that intrinsically limit land use and possibly affect land values. In particular, we investigate if the slope of a land does decrease its price and investigate the role of land slope in forming more reliable constant-quality land price indices and aggregate house price indices. We find that, while land slopes do decrease the land price per unit, they have a small effect on the quality-adjusted land price indices in selected neighborhoods in Auckland, New Zealand, where sloped terrain is common.

**Keywords:** land price index; land slope; slope discount; decomposition; builder's model





## 1. Introduction

Land is one of the most critical inputs in any production function (Chakravorty, 2013 [1]). Its use is possibly the most essential feature that determines urban structure and urban growth, while its value shapes the dynamics of the real estate markets. The land leverage hypothesis states that houses with greater land leverage (i.e., land accounts for a large fraction of the house value) experience a higher price appreciation in a market where there are no increases in the construction cost (Bostic et al., 2007 [2]). For example, a study investigating the Washington, DC metropolitan area from 2000 to 2013 found that variations in land leverage during boom periods notably predicted variations in house prices during bust periods; in addition, land prices were much more volatile than house prices (Davis et al., 2017 [3]).

There are other reasons why the value of land is important, for instance the fact that land value represents a large portion of an individual household's wealth (e.g., Bostic et al., 2007 [2], Bourassa et al. [4]). From a local government's perspective, land value and land-use regulations reciprocally affect each other; such regulations can be related to urban structure (e.g., McMillen & MacDonald, 2002 [5]), urban growth (e.g., Capozza & Helsley, 1989 [6]), or property taxes. Finally, land value is also relevant nationally, being an important part of the National Balance Sheet (e.g., Wentland et al., 2020 [7]). Although land values play such a critical role in the economy, data on them are often difficult to access.

Given the dearth of information on land values, land price is typically measured using one of the following decomposition methods: the vacant land method, the construction cost method, or the hedonic regression method. In the housing prices literature, it has long been acknowledged that housing characteristics should be controlled for in order to maintain a constant quality of the housing price index. Similarly, a well-established fact in the price decomposition literature is that the physical attributes of a house, especially its age, cannot be ignored if one is to obtain a constant-quality price index. However, the literature on the importance of land features is still relatively scarce. Like other price indices, the ideal land price index should represent changes in land prices that are comparable in quality over time.

The importance of geographic features (e.g., proximity to a waterbody, mountains, or wetlands) in urban development and housing supply has generated a growing literature

focused on measuring the role of amenities. For instance, Burchfield et al. (2006) [8] related terrain ruggedness and access to underground water to the density and compactness of new real estate development. Saiz (2010) [9] showed that residential development is considerably constrained by the presence of steep-sloped terrain and found that most areas with inelastic housing supply are severely land-constrained by their topography.

This paper attempts to fill the gap in the decomposition literature by modeling land slope, a proxy for land quality and a factor that possibly discounts land prices, in order to estimate quality-adjusted land price indices. Similar to having constant-quality housing structure price indices, having constant-quality land price indices also requires the properties of the land (e.g., land area and location) to remain constant over time. Of equal importance is the need to take into account the physical attributes of land, especially the land's slope, as these attributes can impose constraints on land development and use. On the one hand, sloping land adds complexity to construction (e.g., extra drainage and extra work in stepping the foundations) and limits land use, hence increasing the construction cost and discounting the land value. On the other hand, sloping land may afford better views, which could increase the property value.

This paper adopts and extends the builder's model (Diewert et al., 2011 [10]) by incorporating the terrain slope to estimate the hedonic pricing of land and to construct constant-quality land price indices; this is done based on data from selected neighborhoods with hilly features from Auckland, New Zealand during the period 2007–2016. Land parcel slopes are prepared in three steps. First, terrain slopes are calculated from the 2013 1-m Digital Elevation Data (DEM) for Auckland. Mean terrain slopes are then calculated for each land parcel extracted from the map of New Zealand Primary Land Parcels. The Address Information Management System (AIMS) from Land Information New Zealand (LINZ) is then used to link land parcels to sales data. In the literature of property appraisal, the residual method of valuation helps the property developers identify a piece of land's re-development value (e.g., Pagourtzi et al., 2003 [11]). However, the residual valuation method requires an estimate on development costs, including the project's construction cost and investor's profit, based on extensive forecasting and many assumptions, making the method susceptible to small changes in the assumptions (Isaac, 1996 [12]). In the builder's model, the use of the exogenous price of the structure per square meter to value the property's structure makes sure that the land value is the residual value of the property transaction value.

Results reveal a slope discount on the price of land per square meter, controlling for land size, land location (i.e., based on school enrollment zone), floor area, age of the house (i.e., in decades), and numbers of rooms. The constant-quality land price indices moderately decrease after controlling for terrain slope, whereas the imputed Fisher chained house price index remained almost unchanged. On the whole, land slope does appear to be an important hedonic characteristic associated with land and hence with house values. However, when the land slope composition does not change over time, having a slope as an additional land characteristic generates minimum effects on the quality-adjusted land price indices.

## 2. Materials and Methods

### 2.1. Four Methods to Compute House Price Indices

There are four primary methods for computing price indices for residential properties: stratification, repeat-sales, appraisal-based methods, and hedonic approaches. More details about computing property price indices can be found in Bailey et al. (1963) [13], Bourassa et al. (2006) [14], Clapp and Giaccotto (1992) [15], De Vries et al. (2009) [16], Wallace and Meese (1997) [17], Wood et al. (2005) [18], and Shiller (1991) [19]. Most recently, Lopez and Hewings (2018) [20] also introduced a method that is based on the repeat-sales (i.e., Case–Shiller) method, while being more flexible; this idea was first suggested by McMillen (2012) [21]. The hedonic regression method is typically the best approach for constructing a constant-quality residential property price index. A typical hedonic estimator

expresses housing prices or their logarithm as a linear function of structural and location attributes. The commonly used hedonic approaches for computing price indices include the hedonic imputation method and the hedonic price method with dummy variables for time.

For the hedonic imputation method, a hedonic regression is initially estimated for each time period separately. For example, consider that there are $N^0$ and $N^1$ houses with $K$ characteristics $z_i^0(z_{i1}^0, z_{i2}^0, \cdots, z_{ik}^0)$ and $z_i^1(z_{i1}^1, z_{i2}^1, \cdots, z_{ik}^1)$ sold in period 0 and period 1, respectively. The following hedonic functions are estimated first:

$$\hat{p}_i^0 = h^0(z_i^0) = \hat{\alpha}^0 + \sum_{k=1}^{K} \hat{\beta}_k^0 \times z_{ik}^0, \tag{1}$$

$$\hat{p}_i^1 = h^1(z_i^1) = \hat{\alpha}^1 + \sum_{k=1}^{K} \hat{\beta}_k^1 \times z_{ik}^1, \tag{2}$$

where $\hat{p}_i^t$ is the predicted sale price of house $i$ sold in period $t$. Next, the change in the quality-controlled house price between two periods is constructed as the price difference between the observed house price in one period and the imputed price; this is only done if the attributes from one period were evaluated at the same estimated prices in the other period. The price of the housing characteristics of period 0, which was imputed in period 1, is denoted as $h^1(z_i^0)$. Similarly, the price of the housing characteristics of period 1, which was imputed in period 0, is denoted as $h^0(z_i^1)$. Holding housing characteristics constant but separate across both period 0 and period 1, we can construct, for example, the following quality-adjusted imputed house price indices:

$$\text{Hedonic Laspeyres Price Index} = \frac{\sum_{i=1}^{N^0} h^1(z_i^0)}{\sum_{i=1}^{N^0} h^0(z_i^0)}, \tag{3}$$

$$\text{Hedonic Paasche Price Index} = \frac{\sum_{i=1}^{N^1} h^1(z_i^1)}{\sum_{i=1}^{N^1} h^0(z_i^1)}. \tag{4}$$

Other important imputed price indices include the Fisher, Geometric-Paasche, Geometric-Laspeyres, and Törnqvist price indices (Hill & Melser, 2008 [22]).

As its name suggests, the hedonic price method with time-dummy variables utilizes cross-sectional data on house prices, which is then expressed in a single equation as a linear combination of time dummies and quality-controlled structural and location attributes. The equation is written as follows:

$$ln(\hat{p}_{i,t}) = \hat{\alpha} + \sum_{t=2}^{T} \hat{\delta}_t D_{i,t} + \sum_{k=1}^{K} \hat{\beta}_k z_{ik,t}, \tag{5}$$

where $D_{i,t}$ represents a set of dummy variables that take on the value of 1 if house $i$ is sold at time $t$ and of 0, otherwise. Moreover, $\hat{\delta}_t$ is interpreted as the quality-adjusted price difference between time $t$ and the baseline time.

A notable problem with the hedonic approaches is that there is often a high correlation between the explanatory variables, which makes the estimated coefficients unstable. As discussed in OECD et al. (2013) [23], multicollinearity is less of a concern if the purpose is to construct an overall constant-quality house price index. However, when the parameters of interest are the coefficients of the physical attributes (e.g., the number of bedrooms) and when the goal is to decompose the overall price index into the land price index and the price index of the housing structure, multicollinearity can be a problem. Schwann (1998) [24] and Diewert et al. (2011, 2015, 2016) [10,25,26] provide a discussion on multicollinearity.

### 2.2. Standard Builder's Model

The builder's model was first discussed by Diewert (2008) [27] and then introduced by Diewert et al. (2011) [10]. It aims to decompose residential price indices into two sub-price indices: a quality-adjusted price index for the housing structure and a price index for the land on which the property is built. This derivation originates from a cost of production approach. From a builder's perspective, the sales price of any property after completion is its expected cost. The expected cost of a property is denoted as the sum of the housing structure cost and the cost of the land on which it is built. The cost of the structure is calculated by multiplying the floor area of the property (e.g., in square meters) by the unit cost of construction (e.g., construction cost per square meter). The cost of the land is calculated by multiplying the land area (e.g., square meters) by the unit cost of land (e.g., cost per square meter). The assumption that the values of land and those of the housing structures are additive is suggested in most of the literature. This includes but is not limited to Bostic et al. (2007) [2], Diewert (2008) [27], Diewert et al. (2011, 2015, 2016) [10,25,26], De Haan and Diewert (2013) [28], and Francke and van de Minne (2017) [29]. Mathematically, the basic builder's model has the following formula:

$$p_{it} = p_t^L L_{it} + p_t^S S_{it} + \varepsilon_{it}, \tag{6}$$

where $p_{it}$ represents the sales price of property $i$ at time $t$; $p_t^L$ and $p_t^S$ are the prices of the land and of the housing structure per square meter at time $t$, respectively; $L_{it}$ is the land area of property $i$ at time $t$; and $S_{it}$ is the floor area of property $i$ at time $t$. The error terms $\varepsilon_{it}$ are assumed to be heteroskedastic, not serially correlated, and mean independent of the covariates.

In essence, the hedonic regression defined in Equation (6) only works for newly built properties. To acknowledge the fact that properties sold at time $t$ include not only newly built properties but also existing older properties, and that older properties are usually worth less than newer properties because of the depreciation of their housing structure over time, Equation (6) is commonly modified by incorporating the age of a property into the baseline builder's model:

$$p_{it} = p_t^L L_{it} + p_t^S (1 - \delta A_{it}) S_{it} + \varepsilon_{it}. \tag{7}$$

Here, $A_{it}$ represents the age of property $i$ at time $t$, while $\delta$ represents the net straight-line deprecation rate as the housing structures of properties age. One can also assume that deprecation rates change over time:

$$p_{it} = p_t^L L_{it} + p_t^S (1 - \delta_t A_{it}) S_{it} + \varepsilon_{it}. \tag{8}$$

Common units of measurement for $A_{it}$ include years and decades. Therefore, $\delta$ can be either the net depreciation rate per year or per decade. Reasonable net annual depreciation rates are in the 0.5–2% range.

If properties are well-maintained or renovated over time, the deterioration of aging properties can be slowed down and, in some cases, older properties may even command a premium. Knight and Sirmans (1996) [30] found that houses with lower-than-average maintenance levels depreciate 0.9% faster per year, while Harding et al. (2007) [31] found that well-maintained houses depreciate 0.5% less per year when compared to the average house. Moreover, older structures can produce functional obstacles (Rubin, 1993 [32]), which can then negatively affect property values. Nevertheless, as housing structures age, some of their aspects may lead to a positive effect on their property value, for example if their design is characteristic to a specific time period. This is recognized as the vintage effect (Coulson & Lahr, 2005 [33]) and can even offset the negative effects of age. For example, Meese and Wallace (1991) [34] found that housing prices increase with age.

Coulson and McMillen (2008) [35] extend the method proposed by McKenzie (2006) [36] to estimate the time, age, and cohort (i.e., vintage) effects simultaneously. Their results show a U-shaped effect of age on housing prices. On the one hand, property prices decrease significantly in the first few years post-construction, while, on the other hand, very old houses have notably high price premia. More recently, Francke and van de Minne (2017) [29] estimated all three age effects on property structures, as well as the time effect on land values. As the builder's model only includes age as a predictor of the housing structure value, $\delta$ should be interpreted as the net effect of age on the structure of a property. Then, $(1 - \delta A_{it})S_{it}$ can be interpreted as being the older structures measured in the units of new or more recent structures. Therefore, due to maintenance information being unavailable, very old structures have been excluded from the model. Burnett-Isaacs et al. (2017) [37] defined old houses as those older than 60 years.

The problem with the straight-line method of modeling depreciation is that the value of the structure can become negative if the structure is old. Therefore, the geometric method is commonly used in national-level research as an alternative to the straight-line method. The builder's model with geometric depreciation has the following form:

$$p_{it} = p_t^L L_{it} + p_t^S (1 - \delta)^{A_{it}} S_{it} + \varepsilon_{it}, \tag{9}$$

where $\delta$ represents the net geometric deprecation rate as the housing structures of properties age. With geometric depreciation, structures deteriorate at a constant rate over time, whereas with a straight-line depreciation structures deteriorate by constant amounts. In practice, empirical studies suggest that it is more appropriate to use the geometric method for residential properties (Chinloy, 1977 [38]; Malpezzi et al., 1987 [39]).

*2.3. Generalization of Standard Builder's Model*

Diewert (2008) [27] suggested that the basic hedonic decomposition can be generalized to incorporate more of the attributes used in the standard hedonic model; this can be done in the following way. Suppose $Z_1, \ldots, Z_M$ are $M$ determinant attributes for the quality of land and $X_1, \ldots, X_H$ are $H$ determinant attributes for the quality of the housing structure; then, the generalized builder's model with geometric depreciation is:

$$p_{it} = p_t^L (1 + \sum_{m=1}^{M} \lambda_m Z_{it,m}) L_{it} + p_t^S (1 - \delta)^{A_{it}} (1 + \sum_{h=1}^{H} \eta_h X_{it,h}) S_{it} + \varepsilon_{it}, \tag{10}$$

where $p_t^L$ is the quality-adjusted price for land at time $t$, and $p_t^S$ is the quality-adjusted price for a housing structure at time $t$. In the literature, location-related attributes are generally used to control for the quality of land. These typically include the distance to the city business center, the zone (e.g., zip code or school zone), and the street pattern of the land on which a property is built, such as if there is an intersection of two streets or a cul-de-sac. Recent work by Pan et al. (2018) [40] suggests that the distance from the Central Business District (CBD) is just one of the many attributes valued by consumers, and hence the land-use changes in a metropolitan region may reflect multiple dimensions of accessibility. Structure characteristics that are controlled for consist of physical attributes such as the number of bathrooms or bedrooms.

For this paper, the school enrollment zone will be incorporated into the model as one of the land characteristics and the numbers of rooms, including both bedrooms and bathrooms, will be used as an additional structural attribute in the generalized model:

$$p_{it} = p_t^L (1 + \sum_{z=1}^{Z} \lambda_z Zone_{it,z}) L_{it} + p_t^S (1 - \delta)^{A_{it}} (1 + \sum_{r=1}^{R} \eta_r Room_{it,r}) S_{it} + \varepsilon_{it}, \tag{11}$$

In this specification, both school zones and numbers of rooms are entered as dummy variables. In addition, to avoid the dummy variable trap, one group from each variable is dropped.

### 2.4. Builder's Model with Terrain Slope

The hedonic literature often adjusts for the quality of housing structures, but there is also a need for quality adjustments when it comes to land characteristics. Cheshire and Sheppard (1995) [41] pointed out that, as land itself is a composite good, land price represents a composite of the price of pure land, the price of the neighborhood and environmental characteristics, and the price of the embodied local public goods.

The theory of land use has its origin in the monocentric city model developed by Alonso (1964) [42], Mills (1967) [43], and Muth (1969) [44]. The traditional monocentric city model treats land as a featureless flat plain, so that locations only differ in their distances to the Central Business District (CBD). Thus, the model predicts that the land prices and the housing density are both higher in those areas closer to the CBD. Later urban economic models extend the monocentric city model to include environmental amenities such as open space (e.g., Anderson & West, 2006 [45]; Geoghegan, 2002 [46]; and Irwin, 2002 [47]) and to allow for multi-centric structures (e.g., Anas & Kim, 1996 [48]; McDonald & McMillen, 1990 [49]; Wieand, 1987 [50]), in order to explain a more complex spatial structure. In addition, more recent literature relaxes the featureless flat plain assumption commonly used in urban economic models. For example, Keenan et al. (2018) [51] developed a conceptual model of what they called climate gentrification and found that price appreciation is positively affected by the incremental increase in elevation in the Miami-Dade County, Florida, which supports the elevation hypothesis. Similarly, Ye and Becker (2017) [52] studied seventeen US cities and found that high-income households prefer to live at higher elevation levels. They also found that the standard deviation of the elevation and that of the relative altitude both positively affect the density and the housing value gradients.

This paper will focus on the terrain slope. If a particular area is flat, then the topography may not influence a house's location and layout; however, on a sloping site, the topography is likely to significantly influence house design. (Flat areas are never strictly horizontal. Instead, they are characterized by gentle slopes that are often hardly noticeable to the naked eye.) Sloping sites present a number of challenges and generally require a greater design input when compared to flat sites. For example, they usually require additional geodetic assessments of slope stability and earthworks before the actual house construction stage. Depending on the steepness of the slope, sloping sites usually have to be cut, filled, and/or retained in order to prepare level plinths on which concrete slab foundations and floors can be laid out.)Increasingly, new houses in New Zealand are built on a concrete slab.) Building on a sloping site may also require additional drainage and sewers. Therefore, the overall construction costs on sloping sites are higher than the overall construction costs on flat sites, which is essentially attributable to the additional amount of cutting, filling, and engineered retaining walls. These costs generally increase with the degree of the slope.

Consequently, in mountainous regions, land slopes might also significantly contribute to the formation of quality-adjusted land prices. Around Auckland, land is visibly uneven, with many houses having been constructed along sloping driveways. If the sample of houses sold in period $t$ consists of more houses on sloping sites than the sample of similarly structured houses sold in previous periods, then changes in topographical characteristics should not be interpreted as changes in land prices over time. If the slope negatively affects housing prices, then it is important to control for this slope; otherwise, the land price index for period $t$ will be underestimated.

We acknowledge that the degree of slope places substantial limitations on the use of land and may add considerable costs to construction due to earthworks projects. Therefore, land slope is modeled as a determinant of land price; other determinants included in the model are land size and the school enrollment zone, which represent the location and the public service associated with a site, respectively. The model can be written as follows:

$$
\begin{aligned}
p_{it} = & \, p_t^L (1 + \sum_{z=1}^{Z} \lambda_z Zone_{it,z})(1 + \sum_{s=1}^{S} \beta_s \, Slope \, Group_{it,s}) L_{it} \\
& + p_t^S (1-\delta)^{A_{it}} (1 + \sum_{r=1}^{R} \eta_r R_{it,r}) S_{it} + \varepsilon_{it},
\end{aligned}
\tag{12}
$$

where $p_t^L$ is the constant-quality land price index (i.e., the "pure" price of land per square meter), and $p_t^S$ is the constant-quality structure price index.

On the one hand, if the ideal site for residential housing is that which provides the desired degree of space at the lowest costs, the difficulty of building on sloping land could mean that the price of a sloping site may be considerably lower than that of a flat site, hence decreasing property values. On the other hand, sloping land may provide better views, hence increasing property values. Due to data limitations, we could not obtain the cost of slope-induced earthwork or the cost of slope-induced superior. Therefore, the estimated coefficient of the slope should be interpreted as the joint effect of these two opposing forces.

The following hypotheses summarize the possible effect of land slope on its hedonic price $\beta_s$:

**Hypothesis 1 (H1).** *If slope-induced construction difficulty has a greater influence than the slope-associated view, then a negative relationship between house price and land slope is expected.*

**Hypothesis 2 (H2).** *If the slope-associated view is more important than the slope-induced difficulty to build, then a positive relationship between house price and land slope is expected.*

**Hypothesis 3 (H3).** *If the slope-induced construction difficulty and the slope-associated view are either equally important or are neither an important house price determinant, then a statistically non-significant relationship between house price and land slope is expected.*

When it comes to computing the quality-adjusted land price indices, the following cases summarize the possible changes in land price indices once we control for land slope:

*Case 1*: If the slope has a negative (positive) effect on the housing price, and if the sample of houses sold in period $t$ consists of more houses built on sloping sites than the sample of similarly structured houses sold in the baseline period, then controlling for slope would adjust the land price index for period $t$ upward (downward).

*Case 2*: If the slope has a negative (positive) effect on the housing price, but the amount of houses sold that were also built on sloping sites does not differ between period $t$ and the baseline period, then controlling for slope would not affect the land price index for period $t$.

*Case 3*: If the slope has no significant effect on the housing price, regardless of the land slope composition over time, then controlling for slope would not affect the land price index.

*2.5. Data*

The analysis in this paper relies on a data set that combines information on housing sales and on land parcel slopes of sold houses.

2.5.1. Sales Data

Property-level sales data on three Auckland school enrollment zones between January 2007 and December 2016 was purchased from Quotable Value Limited (QV), powered by CoreLogic NZ Ltd, an entity responsible for conducting property market valuations in New Zealand. The data encompasses the Double Grammar Zone (i.e., an overlapping area of enrollment zones of the Auckland Grammar School and Epsom Girls' Grammar School), the Selwyn College zone, and the One Tree Hill College zone.

Sales data contain information on sales price, sales date, property address, and a set of structural property attributes. The analysis is targeted at all types of houses, but it excludes apartments. We include residential properties with fully detached or semi-detached houses that are situated on their own clearly defined piece of land, while removing those units with missing or misreported information. Outliers for sales price, land area, floor area, bedrooms, and bathrooms are dropped by year of sale within each school zone. First, the bottom 1% and the top 3% of sales prices are dropped. Then, the top 1% of land areas are trimmed, followed by the top 1% of floor areas. The data set is further filtered by eliminating those observations with the number of bathrooms and the number of bedrooms being in the top 1%. We also exclude houses that were built before the 1950s. The final sample contains 5657 observations for the period 2007–2016.

Two land characteristics of the sales data used in this analysis are the land area ($m^2$) and the school enrollment zone in which the land is located. Structural characteristics used in the analysis include the building's age and its floor area ($m^2$). The original age of the building is coded in decade-long construction periods, such as the 1940s and the 2010s. Following Diewert et al. (2015) [25], the original age of the building is recoded into decade age using the following procedure: the most recent construction period for any of the houses sold between 2007 and 2009 was the 2000s. Hence, the age variable for the construction decade is calculated as (2000–construction period)/10. From 2010 onward, the newest houses sold were built in the 2010s. Hence, the corresponding age variable is calculated as (2010–construction period)/10. After this recoding procedure, a house built in the 2000s and sold between 2007 and 2009 has a decade age of 0, whereas a house built in the 2000s that was sold in 2010s has a decade age of 1. Table 1 presents the descriptive statistics for the sample of interest. On average, the houses in the sample were built two decades ago. The sample's mean sales price is 1,164,640 NZ dollars (NZ$), with an average land and floor area of about 580 $m^2$ and 217 $m^2$, respectively.

**Table 1.** Descriptive statistics.

|  | **Mean** | **Std. Dev.** | **Min** | **Max** |
|---|---|---|---|---|
| Sales Price (1000 NZ$) | 1164.64 | 695.15 | 300.00 | 5880.00 |
| House Age (decades) | 2.37 | 2.19 | 0.00 | 6.00 |
| Land Area ($m^2$) | 580.53 | 256.30 | 116.00 | 2048.00 |
| Floor Area ($m^2$) | 216.65 | 79.61 | 43.00 | 530.00 |
| Rooms | 5.778 | 1.44 | 2.00 | 5.00 |
| Land Slope (%) | 18.55 | 12.05 | 1.53 | 69.57 |
| Number of Observations | | 5657 | | |

Note: This table presents the descriptive statistics for the selected neighborhoods in Auckland, New Zealand, from 2007 to 2016. Land slope is measured as the percentage of the increase (%).

2.5.2. Computing the Land Slope

The land slope used in this paper is obtained from a light detection and ranging (LiDAR) 1-meter resolution digital elevation model (DEM) fitted to the map of New Zealand Primary Land Parcels using ArcGIS. Both maps are converted to the New Zealand Transverse Mercator 2000 (NZTM2000) projection for analysis.

The airborne Auckland LiDAR 1m DEM data was captured in 2013 for the Auckland Council by NZ Aerial Mapping & Aerial Surveying Limited. It was collected at a point density of more than 1.5 points per square meter. The 1 m DEM data was downloaded from the Land Information New Zealand (LINZ) Data Service. More information about DEM can be found at https://data.linz.govt.nz/layer/53405-auckland-lidar-1m-dem-2013/ accessed on 9 February 2019. Elevation values are measured in meters. In ArcGIS, the unit of measure for the z (elevation) unit is also the meter, so the z-factor of value 1 is used to calculate the percentage by which the values of slopes rise in each DEM cell (i.e., the rate of change in elevation). (There are two options for the units of measurement for terrain slope: degree values and percentage elevation values. Please see Appendix A for more information.)

The map of the New Zealand Primary Land Parcels was also downloaded using the LINZ Data service. More information about NZ Primary Parcels can be found at https://data.linz.govt.nz/layer/50772-nz-primary-parcels/data/ accessed on 11 February 2019. To determine the terrain slope of each land parcel, the Zonal Statistic tools in ArcGIS were used. Each land parcel on the land parcel map was treated as an input zone, and the parcel ID was used to define the zones. The raster created from the 1 m DEM contains the slope values and is then used to calculate each parcel's mean slope. The resulting map of parcel slopes is depicted in Figure 1. For reference, an aerial map of the study area is shown in Figure 2. Table 1 reports that the average slope of the sample is 18.55% or 10.51°. The slopes are then divided into six broad groups according to the slope classes from the Land Resource Information System (LRIS). (Please see Appendix A Table A1 for a range of slope classifications from different countries.) As presented in Table 2, the groups to be used in the analysis are: flat to gently undulating (0–3°), undulating (4–7°), rolling (8–15°), strongly rolling (16–20°), moderately steep (21–25°), and steep (26–35°). (Observations with a slope of more than 35°were also excluded from the final sample.) Table 3 shows that 41.7% of the sample is in the rolling slope range. The correlations of the land slope with the land area, floor area, and the total number of rooms are 0.2285, 0.2148, and 0.1462, respectively.

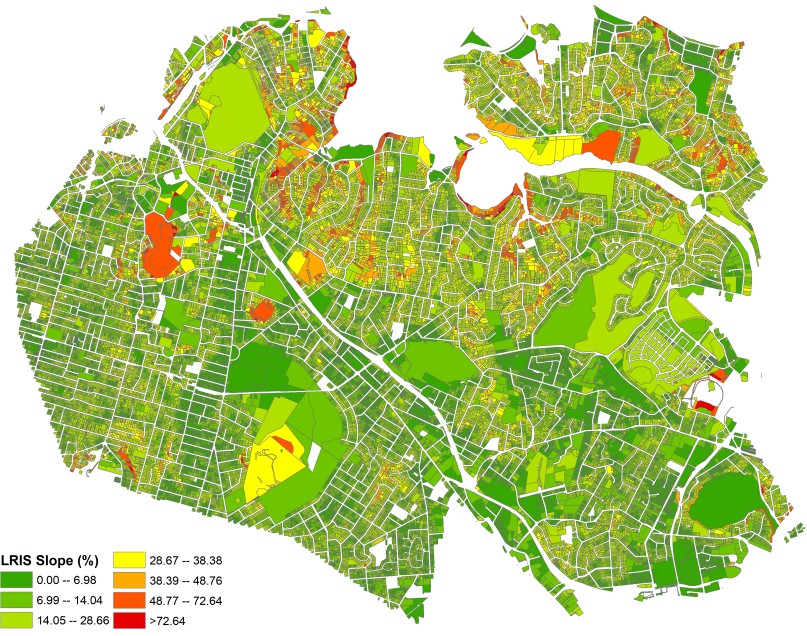

**Figure 1.** Land Parcel Slope in Percentages. This figure is produced by the author in ArcGIS using 1 m DEM fitted to the map of primary land parcels covering the study area.

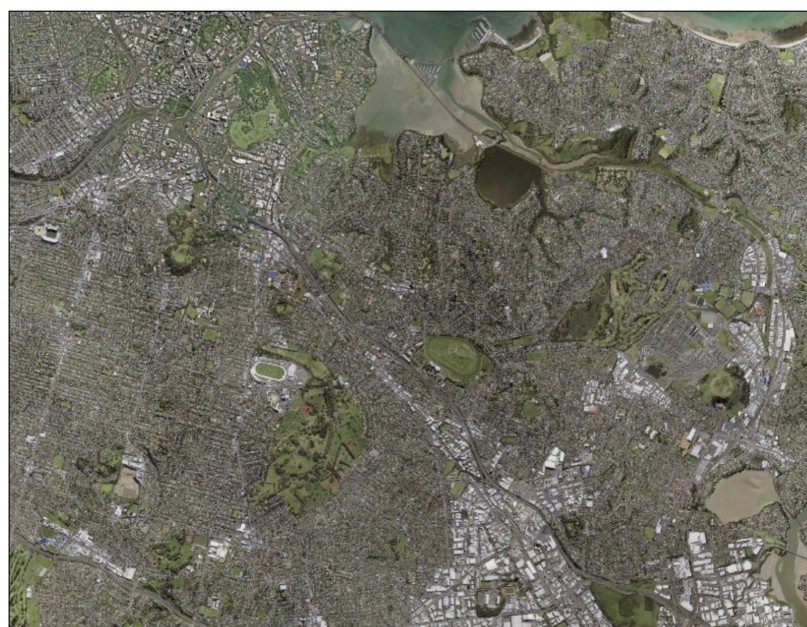

**Figure 2.** Aerial photo reference. Source: Land Information New Zealand access from ArcGIS.

**Table 2.** New Zealand LRIS Slope Classes.

| Slope Classes | Degree° | Percent Rise (%) |
|---|---|---|
| Flat to gently undulating | 0–3 | 0–5.24 |
| Undulating | 4–7 | 6.99–12.28 |
| Rolling | 8–15 | 14.05–26.79 |
| Strongly rolling | 16–20 | 28.67–36.40 |
| Moderately steep | 21–25 | 38.39–46.63 |
| Steep | 26–35 | 48.77–70.02 |
| Very steep | 36–42 | 72.65–90.04 |
| Precipitous | >42 | >90.04 |

Note: This table presents the LRIS slope classes, which are accessible at https://lris.scinfo.org.nz/document/9162-lris-data-dictionary-v3/ accessed on 1 March 2019.

**Table 3.** Frequency of slope classes.

| LRIS Slope Classes | Selwyn | One Tree Hill | Double Grammar | Total |
|---|---|---|---|---|
| Flat to gently undulating (0–3°) | 7.01 | 21.14 | 11.49 | 12.21 |
| Undulating (4–7°) | 22.21 | 37.96 | 21.54 | 26.48 |
| Rolling (8–15°) | 46.52 | 34.83 | 40.93 | 41.70 |
| Strongly rolling (16–20°) | 14.53 | 4.63 | 12.79 | 11.26 |
| Moderately steep (21–25°) | 6.29 | 1.13 | 8.22 | 5.36 |
| Steep (26–35°) | 3.44 | 0.31 | 5.03 | 2.99 |
| Total | 100 | 100 | 100 | 100 |
| Number of Observations | | 5657 | | |

Note: This table presents the frequency of the LRIS slope classes within each school enrollment zone included in our study area.

### 2.5.3. Linking Parcels with Addresses

To link the computed land parcel slopes to sales data, the Address Component data from the LINZ's Address Information Management System (AIMS) is used. (More information about the AIMS Address Component data can be found at https://data.linz.govt.nz/table/53354-aims-address-component/data/ accessed on 1 March 2019. AIMS Address Component data contains information on address ID, parcel ID, and on the components of each address, such as address number, street number, and road name. Parcel ID is used to link the mean slope data to the AIMS data. Address components are combined to a

single full address based on the address component order; this address is then linked to the sales data.

## 3. Empirical Results

### 3.1. Exogenous Information on the Prices of Housing Structures

The practical problem with the models defined by Equations (7)–(12) is that the multicollinearity between the land area and the housing structure area can result in highly unstable and unreasonable estimates. Empirical evidence (e.g., Diewert et al. 2011, 2015, 2016 [10,25,26]) suggests that an approach using exogenous information on the price of housing structures can overcome this multicollinearity problem and produce more reasonable and stable price dynamics for both land and structure. Such exogenous prices are usually the new construction price indices reported by a statistical agency. (We also estimated models with a straight-line depreciation and models where the price of new housing structures grows proportionally to the exogenous construction cost index and at a constant rate (i.e., $p_s^t = \theta \gamma_t$). Nevertheless, these models' results were not satisfactory, which directly confirms the need to use geometric depreciation and the use of the exogenous construction cost index (i.e., $p_s^t = \gamma_t$).) As indicated by Rosenthal (1999) [53], the long-run equilibrium price of new structures equals the current construction costs.

The quarterly housing construction cost index is derived from the building consent statistics for new homes in the Auckland region, which were obtained from Statistics New Zealand (Stats NZ). Building consent statistics contain information about the numbers, values, and floor areas of new homes or non-residential buildings, and about the alterations that were approved for construction. More information about building consent can be found at https://www.stats.govt.nz/information-releases/building-consents-issued-may-2018 accessed on 16 March 2019. To compute the quarterly housing construction cost index $\gamma_t$, the actual values of the new homes approved for construction in quarter $t$ are divided by the floor areas of the new homes approved for construction in the same quarter:

$$\gamma_t = \frac{\text{value of new homes approved for construction}_t}{\text{floor area of new homes approved for construction}_t}. \tag{13}$$

The quarterly building consent statistics for new houses in the Auckland region are presented in Table 4, with the quarterly construction indices calculated using Equation (13) being reported in column 3; it appears that the construction cost per square meter increased by about 56.76% from 2007 to 2016. These values are not inflation-adjusted.

### 3.2. Results from the Builder's Models

The builder's models to be estimated are all nonlinear models and are estimated using iteration methods that require starting values for the parameters. To facilitate the convergence of the estimation algorithm for models with more parameters, estimates from the models with fewer parameters will be used as the starting values in the estimation of models with more parameters.

Instead of estimating 40 standard builder's models as defined in Equation (9), namely one for each quarter from the first quarter of 2007 to the fourth quarter of 2016, the combined version of Equation (9) is estimated using 40 quarterly dummy variables. The combined estimation allows for the comparison of log-likelihood values across models. In the combined standard model, there are three explanatory variables (i.e., land area, floor area, and decade house age) and 41 parameters (i.e., 40 quarterly land prices and the net decade depreciation rate $\delta$) to be estimated.

**Table 4.** Quarterly building consent for new houses—Auckland region.

| Year | Quarter | Value NZ$ | Floor Area m$^2$ | Construction Index NZ$/m$^2$ |
|------|---------|-----------|------------------|------------------------------|
|      |         | (1)       | (2)              | (3)                          |
| 2007 | Q1 | 308,470,997 | 252,573 | 1221 |
| 2007 | Q2 | 300,770,306 | 239,233 | 1257 |
| 2007 | Q3 | 345,485,007 | 273,549 | 1263 |
| 2007 | Q4 | 351,445,145 | 272,177 | 1291 |
| 2008 | Q1 | 286,205,722 | 218,038 | 1313 |
| 2008 | Q2 | 246,163,735 | 180,682 | 1362 |
| 2008 | Q3 | 198,047,620 | 147,618 | 1342 |
| 2008 | Q4 | 174,633,586 | 119,527 | 1461 |
| 2009 | Q1 | 168,884,517 | 116,789 | 1446 |
| 2009 | Q2 | 168,796,144 | 112,015 | 1507 |
| 2009 | Q3 | 211,740,103 | 144,729 | 1463 |
| 2009 | Q4 | 265,322,840 | 182,055 | 1457 |
| 2010 | Q1 | 242,630,596 | 172,010 | 1411 |
| 2010 | Q2 | 278,702,453 | 200,389 | 1391 |
| 2010 | Q3 | 244,406,195 | 173,977 | 1405 |
| 2010 | Q4 | 219,308,410 | 154,478 | 1420 |
| 2011 | Q1 | 224,515,030 | 151,485 | 1482 |
| 2011 | Q2 | 216,079,500 | 146,956 | 1470 |
| 2011 | Q3 | 253,871,795 | 172,412 | 1472 |
| 2011 | Q4 | 289,341,990 | 196,760 | 1471 |
| 2012 | Q1 | 288,826,634 | 191,250 | 1510 |
| 2012 | Q2 | 302,402,403 | 195,794 | 1544 |
| 2012 | Q3 | 291,687,676 | 192,531 | 1515 |
| 2012 | Q4 | 374,062,098 | 248,169 | 1507 |
| 2013 | Q1 | 349,495,817 | 231,893 | 1507 |
| 2013 | Q2 | 416,242,630 | 263,492 | 1580 |
| 2013 | Q3 | 418,365,595 | 268,079 | 1561 |
| 2013 | Q4 | 424,108,493 | 257,157 | 1649 |
| 2014 | Q1 | 450,361,701 | 280,527 | 1605 |
| 2014 | Q2 | 462,789,605 | 275,864 | 1678 |
| 2014 | Q3 | 455,226,457 | 270,563 | 1683 |
| 2014 | Q4 | 498,182,013 | 288,158 | 1729 |
| 2015 | Q1 | 450,550,041 | 257,013 | 1753 |
| 2015 | Q2 | 505,478,008 | 291,640 | 1733 |
| 2015 | Q3 | 553,545,632 | 318,161 | 1740 |
| 2015 | Q4 | 611,276,364 | 334,010 | 1830 |
| 2016 | Q1 | 599,102,433 | 312,367 | 1918 |
| 2016 | Q2 | 676,609,794 | 364,616 | 1856 |
| 2016 | Q3 | 672,485,586 | 353,101 | 1905 |
| 2016 | Q4 | 575,858,608 | 300,845 | 1914 |

Note: The quarterly housing construction cost index is derived from the building consent statistics for new homes in the Auckland region, which were obtained from Statistics New Zealand (Stats NZ).

Estimation results are reported in column 1 of Table 5. The adjusted R-squared shows that the three-predictor nonlinear model explains 86.2% of the variation in sales prices. The 40 estimated quarterly land prices show that the land price increased 2.40-fold over the 10 years of interest (see normalized values in column 2 of Table 6), which is a much greater rate than the 1.57-fold increase in construction cost index over the same period (see normalized values in column 1 of Table 6). The estimated net decade depreciation rate $\delta$ is 0.074 or 7.4% per decade; this corresponds to an net annual depreciation rate of 0.74% per year and is comparable to the net annual depreciation rates of the standard models reported by Diewert et al. (2016) [26] and the net annual depreciation rates of single-family owner-occupied housing reported by Chinloy (1977) [38]. (Diewert et al. (2016) [26] suggested that the net annual geometric depreciation rate is between 1 and 4%. For London, Chinloy (1977) [38] estimated the net annual geometric rate of single-family, owner-occupied housing to be between 0.69 and 0.91%.)

Using the estimated coefficients from the standard model as the initial values, the combined version of the generalized builder's model defined in Equation (11) is estimated, with the inclusion of school zones and room numbers in the model. To avoid the dummy variable trap, the Selwyn College school zone and the category of houses with 2–4 rooms serve as the reference groups for the two variables, respectively. (As only 4.55% (n = 419) of the observations were houses with two or three rooms, these were re-grouped together with the four-room houses. The 2- to 4-room group is set as the baseline group.) This nonlinear model consists of five explanatory variables (i.e., land area, floor area, decade house age, school zone, number of rooms) and 47 parameters to be estimated.

Estimated results are presented in column 2 of Table 5. The adjusted R-squared shows that the five-predictor model explains 93.8% of the variation in sales prices. Moreover, the log-likelihood, AIC, and BIC all indicate that including two school zones and four room categories as explanatory variables leads to a statistically significant improvement in the model fit when compared to the standard model. After controlling for additional structural and land characteristics, the estimated quarterly land prices point to a 2.82-fold increase in land price over the 10 years of interest (see normalized values in column 3 of Table 6), thus higher than the 2.40-fold increase observed in the standard model. In addition, the net decade depreciation rate is now estimated at 6.4%, which corresponds to a net annual depreciation rate of 0.64%. All else being equal, when compared to houses that have between two and four rooms, it costs about NZ$1100 more per square meter to build a five- or a six-room house, and about NZ$1300 more per square meter to build a house with more than seven rooms. This finding is reasonable, since more building materials and a longer construction time are required when building houses with more rooms. As a result, both the costs of material and that of labor will increase with each increase in the number of rooms. This model also shows that, when compared to the baseline Selwyn College zone, it is on average about NZ$360 per square meter cheaper to reside in the One Tree Hill College zone and about NZ$552 per square meter more expensive to reside in the Double Grammar Zone; this is consistent with market observations. The Double Grammar Zone is the most sought-after state school zone in Auckland, with mean property values that are constantly reported to be hundreds of thousands of dollars higher than outside this specific enrollment zone. In addition, the number of enrollments in both the Auckland Grammar School and the Epsom Girls' Grammar School has approached its maximum values, due to the increase in school-age residents in the Double Grammar Zone. Together, the high demand for and almost saturated supply of places in the two prestigious schools have driven up property prices in the area. Therefore, the particularly high estimated land price in the Double Grammar Zone can be seen as a financial premium and can be attributed to the increasing demand for and shortage of land within that zone.

Following this, we turn to the generalized model with land slope and estimate the combined version of the model defined in Equation (12). Estimated coefficients from the generalized model without land slope were used as the starting values. The rolling slope category is set at the reference land slope category because 41.7% of the observations belong to this range.

Results for the 52 parameters of the six-predictor nonlinear model are presented in column 3 of Table 5. The adjusted R-squared increases slightly to 0.940. Nonetheless, the log-likelihood, AIC, and BIC all indicate that adding five site-slope parameters to the model does result in a statistically significant improvement in model fit when compared to the generalized model without terrain slopes. After controlling for land slope, the quarterly constant-quality land indices that are estimated show that land prices increased 2.78-fold over the 10 years of interest (see normalized values in column 4 of Table 6), which is a lower increase when compared to the 2.82-fold increase obtained from the previous model. The results for the net decade depreciation rate, school zones, and number of rooms are consistent with previous estimates.

**Table 5.** Builder's models—estimation results.

| | Standard (1) | | Generalized w/o Slope (2) | | Generalized w/ Slope (3) | |
|---|---|---|---|---|---|---|
| | Coef. | Std.Err. | Coef. | Std.Err. | Coef. | Std.Err. |
| 2007Q1 | 0.980 *** | (0.060) | 0.653 *** | (0.051) | 0.692 *** | (0.054) |
| 2007Q2 | 1.104 *** | (0.065) | 0.687 *** | (0.061) | 0.748 *** | (0.052) |
| 2007Q3 | 1.182 *** | (0.094) | 0.707 *** | (0.065) | 0.731 *** | (0.068) |
| 2007Q4 | 0.987 *** | (0.063) | 0.625 *** | (0.049) | 0.643 *** | (0.050) |
| 2008Q1 | 1.030 *** | (0.090) | 0.661 *** | (0.051) | 0.688 *** | (0.054) |
| 2008Q2 | 0.931 *** | (0.079) | 0.546 *** | (0.043) | 0.579 *** | (0.041) |
| 2008Q3 | 1.043 *** | (0.104) | 0.598 *** | (0.085) | 0.631 *** | (0.090) |
| 2008Q4 | 1.057 *** | (0.078) | 0.495 *** | (0.053) | 0.526 *** | (0.059) |
| 2009Q1 | 0.913 *** | (0.068) | 0.420 *** | (0.050) | 0.446 *** | (0.054) |
| 2009Q2 | 1.040 *** | (0.065) | 0.480 *** | (0.047) | 0.501 *** | (0.047) |
| 2009Q3 | 0.999 *** | (0.060) | 0.520 *** | (0.044) | 0.569 *** | (0.041) |
| 2009Q4 | 1.082 *** | (0.062) | 0.590 *** | (0.042) | 0.637 *** | (0.044) |
| 2010Q1 | 0.949 *** | (0.061) | 0.559 *** | (0.040) | 0.592 *** | (0.040) |
| 2010Q2 | 1.140 *** | (0.074) | 0.680 *** | (0.055) | 0.716 *** | (0.058) |
| 2010Q3 | 1.231 *** | (0.086) | 0.648 *** | (0.060) | 0.669 *** | (0.064) |
| 2010Q4 | 1.056 *** | (0.074) | 0.594 *** | (0.057) | 0.625 *** | (0.058) |
| 2011Q1 | 1.051 *** | (0.076) | 0.528 *** | (0.055) | 0.568 *** | (0.053) |
| 2011Q2 | 1.175 *** | (0.078) | 0.684 *** | (0.064) | 0.713 *** | (0.065) |
| 2011Q3 | 1.103 *** | (0.074) | 0.562 *** | (0.077) | 0.617 *** | (0.064) |
| 2011Q4 | 1.117 *** | (0.076) | 0.654 *** | (0.067) | 0.689 *** | (0.067) |
| 2012Q1 | 1.179 *** | (0.071) | 0.655 *** | (0.049) | 0.688 *** | (0.049) |
| 2012Q2 | 1.128 *** | (0.055) | 0.616 *** | (0.044) | 0.660 *** | (0.043) |
| 2012Q3 | 1.169 *** | (0.057) | 0.647 *** | (0.042) | 0.682 *** | (0.046) |
| 2012Q4 | 1.359 *** | (0.072) | 0.813 *** | (0.042) | 0.847 *** | (0.042) |
| 2013Q1 | 1.241 *** | (0.064) | 0.747 *** | (0.044) | 0.796 *** | (0.044) |
| 2013Q2 | 1.536 *** | (0.076) | 0.910 *** | (0.050) | 0.953 *** | (0.054) |
| 2013Q3 | 1.541 *** | (0.083) | 0.966 *** | (0.058) | 1.025 *** | (0.060) |
| 2013Q4 | 1.504 *** | (0.088) | 0.908 *** | (0.061) | 0.982 *** | (0.060) |
| 2014Q1 | 1.552 *** | (0.077) | 0.928 *** | (0.060) | 0.962 *** | (0.062) |
| 2014Q2 | 1.769 *** | (0.102) | 1.138 *** | (0.076) | 1.174 *** | (0.080) |
| 2014Q3 | 1.873 *** | (0.100) | 1.219 *** | (0.076) | 1.261 *** | (0.079) |
| 2014Q4 | 1.886 *** | (0.080) | 1.164 *** | (0.055) | 1.218 *** | (0.059) |
| 2015Q1 | 2.049 *** | (0.122) | 1.409 *** | (0.083) | 1.508 *** | (0.082) |
| 2015Q2 | 2.000 *** | (0.097) | 1.403 *** | (0.071) | 1.471 *** | (0.072) |
| 2015Q3 | 2.021 *** | (0.095) | 1.451 *** | (0.066) | 1.524 *** | (0.066) |
| 2015Q4 | 2.041 *** | (0.096) | 1.338 *** | (0.064) | 1.423 *** | (0.060) |
| 2016Q1 | 2.360 *** | (0.142) | 1.603 *** | (0.091) | 1.660 *** | (0.093) |
| 2016Q2 | 2.203 *** | (0.108) | 1.536 *** | (0.082) | 1.637 *** | (0.073) |
| 2016Q3 | 2.285 *** | (0.106) | 1.612 *** | (0.085) | 1.669 *** | (0.085) |
| 2016Q4 | 2.350 *** | (0.180) | 1.842 *** | (0.143) | 1.922 *** | (0.137) |
| Decade Discount Rate $\delta$ | 0.074 *** | (0.020) | 0.064 *** | (0.007) | 0.066 *** | (0.007) |
| One Tree Hill School Zone | | | −0.360 *** | (0.015) | −0.398 *** | (0.014) |
| Double Grammar Zone | | | 0.552 *** | (0.039) | 0.536 *** | (0.036) |
| 5 Rooms | | | 1.083 *** | (0.041) | 1.036 *** | (0.042) |
| 6 Rooms | | | 1.092 *** | (0.043) | 1.061 *** | (0.043) |
| 7 Rooms | | | 1.288 *** | (0.046) | 1.262 *** | (0.046) |
| 8+ Rooms | | | 1.281 *** | (0.046) | 1.230 *** | (0.047) |
| Flat to gently undulating (0–3°) | | | | | 0.117 *** | (0.037) |
| Undulating (4–7°) | | | | | 0.042 | (0.027) |
| Strongly rolling (16–20°) | | | | | −0.038 | (0.036) |
| Moderately steep (21–25°) | | | | | −0.168 *** | (0.035) |
| Steep (26–35°) | | | | | −0.268 *** | (0.044) |
| Adjusted $R^2$ | 0.862 | | 0.938 | | 0.940 | |
| Log-Likelihood | −43,198.376 | | −40,949.968 | | −40,852.527 | |
| AIC | 86,478.751 | | 81,993.935 | | 81,809.053 | |
| BIC | 86,751.018 | | 82,306.046 | | 82,154.367 | |
| Number of Observations | 5657 | | 5657 | | 5657 | |

Note: This table presents the estimation results for the three builder's models. The Selwyn College school zone is used as the baseline school zone. The reference room group is that of houses with 2–4 rooms. Rolling land (8–15°) is the baseline land slope class. Robust standard errors are reported in parentheses. * $p < 0.10$, ** $p < 0.05$, *** $p < 0.01$.

**Table 6.** Constant-quality sub-price indices and aggregate house price indices.

| Quarter | Structure Price Indices | Land Price Indices | | | Fisher Chained House Price Indices | | | Hedonic House Price Indices |
| | | Standard | Generalized | | Standard | Generalized | | |
| | | | w/o Slope | w/ Slope | | w/o Slope | w/ Slope | |
| | (1) | (2) | (3) | (4) | (5) | (6) | (7) | (8) |
|---|---|---|---|---|---|---|---|---|
| 2007Q1 | 1 | 1 | 1 | 1 | 1 | 1 | 1 | 1 |
| 2007Q2 | 1.0295 | 1.1269 | 1.0528 | 1.0809 | 1.1004 | 1.0407 | 1.0546 | 1.0757 |
| 2007Q3 | 1.0344 | 1.2062 | 1.0835 | 1.0564 | 1.1582 | 1.0576 | 1.0456 | 1.0525 |
| 2007Q4 | 1.0573 | 1.0072 | 0.9576 | 0.9294 | 1.0233 | 1.0113 | 0.9972 | 1.0391 |
| 2008Q1 | 1.0753 | 1.0513 | 1.0137 | 0.9939 | 1.0603 | 1.0476 | 1.0381 | 1.0521 |
| 2008Q2 | 1.1155 | 0.9500 | 0.8365 | 0.8360 | 0.9970 | 0.9801 | 0.9780 | 0.9889 |
| 2008Q3 | 1.0991 | 1.0644 | 0.9171 | 0.9116 | 1.0730 | 1.0077 | 1.0041 | 0.9739 |
| 2008Q4 | 1.1966 | 1.0786 | 0.7592 | 0.7603 | 1.1135 | 0.9985 | 0.9956 | 0.9912 |
| 2009Q1 | 1.1843 | 0.9314 | 0.6442 | 0.6440 | 1.0078 | 0.9424 | 0.9381 | 0.9535 |
| 2009Q2 | 1.2342 | 1.0620 | 0.7354 | 0.7234 | 1.1136 | 1.0098 | 1.0008 | 1.0093 |
| 2009Q3 | 1.1982 | 1.0197 | 0.7963 | 0.8227 | 1.0733 | 1.0173 | 1.0266 | 1.0347 |
| 2009Q4 | 1.1933 | 1.1048 | 0.9048 | 0.9197 | 1.1323 | 1.0652 | 1.0695 | 1.0551 |
| 2010Q1 | 1.1556 | 0.9688 | 0.8571 | 0.8550 | 1.0232 | 1.0221 | 1.0179 | 1.0296 |
| 2010Q2 | 1.1392 | 1.1631 | 1.0418 | 1.0338 | 1.1598 | 1.1047 | 1.0995 | 1.0645 |
| 2010Q3 | 1.1507 | 1.2562 | 0.9924 | 0.9669 | 1.2292 | 1.0877 | 1.0735 | 1.0747 |
| 2010Q4 | 1.1630 | 1.0780 | 0.9106 | 0.9035 | 1.1049 | 1.055 | 1.0489 | 1.0241 |
| 2011Q1 | 1.2138 | 1.0730 | 0.8095 | 0.8212 | 1.1160 | 1.0348 | 1.0364 | 1.0034 |
| 2011Q2 | 1.2039 | 1.1997 | 1.0477 | 1.0295 | 1.2012 | 1.1340 | 1.1243 | 1.0754 |
| 2011Q3 | 1.2056 | 1.1253 | 0.8615 | 0.8918 | 1.1491 | 1.0506 | 1.0614 | 1.0877 |
| 2011Q4 | 1.2048 | 1.1397 | 1.0028 | 0.9953 | 1.1592 | 1.1172 | 1.1112 | 1.0920 |
| 2012Q1 | 1.2367 | 1.2038 | 1.0043 | 0.9935 | 1.2139 | 1.1353 | 1.1274 | 1.1388 |
| 2012Q2 | 1.2645 | 1.1517 | 0.9435 | 0.9533 | 1.1856 | 1.1227 | 1.1236 | 1.1491 |
| 2012Q3 | 1.2408 | 1.1936 | 0.9918 | 0.9853 | 1.2086 | 1.1329 | 1.1266 | 1.1869 |
| 2012Q4 | 1.2342 | 1.3874 | 1.2457 | 1.2242 | 1.3451 | 1.2482 | 1.2373 | 1.2284 |
| 2013Q1 | 1.2342 | 1.2670 | 1.1442 | 1.1500 | 1.2600 | 1.2027 | 1.2034 | 1.2492 |
| 2013Q2 | 1.2940 | 1.5681 | 1.3941 | 1.3769 | 1.4887 | 1.3464 | 1.3381 | 1.3678 |
| 2013Q3 | 1.2785 | 1.5734 | 1.4799 | 1.4810 | 1.4878 | 1.3773 | 1.3782 | 1.3842 |
| 2013Q4 | 1.3505 | 1.5351 | 1.3909 | 1.4184 | 1.4810 | 1.3745 | 1.3869 | 1.4249 |
| 2014Q1 | 1.3145 | 1.5845 | 1.4226 | 1.3903 | 1.5058 | 1.3698 | 1.3545 | 1.4449 |
| 2014Q2 | 1.3743 | 1.8057 | 1.7444 | 1.6965 | 1.6808 | 1.5538 | 1.5338 | 1.5435 |
| 2014Q3 | 1.3784 | 1.9115 | 1.8683 | 1.8212 | 1.7568 | 1.6133 | 1.595 | 1.5527 |
| 2014Q4 | 1.4161 | 1.9249 | 1.7844 | 1.7596 | 1.7775 | 1.5952 | 1.5863 | 1.6149 |
| 2015Q1 | 1.4357 | 2.0916 | 2.1599 | 2.1785 | 1.9017 | 1.7835 | 1.798 | 1.8099 |
| 2015Q2 | 1.4193 | 2.0416 | 2.1494 | 2.1248 | 1.8613 | 1.7699 | 1.7637 | 1.8664 |
| 2015Q3 | 1.4251 | 2.0628 | 2.2243 | 2.2024 | 1.8782 | 1.8085 | 1.8041 | 1.9114 |
| 2015Q4 | 1.4988 | 2.0833 | 2.0503 | 2.0566 | 1.9125 | 1.7617 | 1.7692 | 1.8847 |
| 2016Q1 | 1.5708 | 2.4089 | 2.4568 | 2.3981 | 2.1669 | 1.9936 | 1.9724 | 2.0373 |
| 2016Q2 | 1.5201 | 2.2481 | 2.3541 | 2.3655 | 2.0373 | 1.9182 | 1.9301 | 2.1294 |
| 2016Q3 | 1.5602 | 2.3325 | 2.4709 | 2.4115 | 2.1093 | 1.9953 | 1.9729 | 2.1108 |
| 2016Q4 | 1.5676 | 2.3984 | 2.8225 | 2.7768 | 2.1593 | 2.1703 | 2.1585 | 2.1618 |

Note: This table reports the normalized land price indices and the imputed Fisher chained house price indices for each of the builder's models, as well as the housing structures price indices and the hedonic house price indices.

These results also show that land price per square meter decreases with each increase in land slope, which indicates that the difficulty of building on sloping land has a greater influence on pricing than the possibly superior slope-associated views; this offers support for our first hypothesis. In addition, flat to gently undulating land (0–3°) is, on average, NZ$117 per square meter more expensive than rolling land (8–15°). The small positive price difference between undulating (4–7°) and rolling land, and the small negative price difference between strongly rolling (16–20°) and rolling land are statistically non-significant. In contrast, moderately steep (21–25°) and steep (26–35°) land are cheaper by NZ$168 and NZ$268 per square meter, respectively, when compared to rolling land. These results support the theory outlined in the previous sections, namely that the difficulty and complexity associated with building on steeper land lead to lower land prices. Estimation results using alternative land slope classifications are presented in Table A2 in the Appendix A and are consistent with the main results.

### 3.3. Construction of the Overall House Price Index

Builder's models decompose the sales price into the constant-quality price of land and the constant-quality price of housing structures; following several steps, these can be combined to generate an overall house price index. First, utilizing the estimates from the generalized builder's model with land slopes, we can construct the imputed constant-quality

amount of land ($\widehat{IL}_{it}$)) and the imputed constant-quality amount of housing structures $\widehat{IS}_{it}$)) for each house $i$ sold in quarter $t$ as follows:

$$\widehat{IL}_{it} = (1 + \sum_{z=1}^{Z} \widehat{\lambda}_z Zone_{it,z})(1 + \sum_{s=1}^{S} \widehat{\beta}_s \; Slope \; Group_{it,s})L_{it}, \tag{14}$$

$$\widehat{IS}_{it} = (1 - \widehat{\delta})^{A_{it}}(1 + \sum_{r=1}^{R} \widehat{\eta}_r R_{it,r})S_{it}. \tag{15}$$

Then, the total constant-quality amount of land $\widehat{IL}_t$ and the total constant-quality amount of housing structures $\widehat{IS}_t$ in quarter $t$ can be computed by aggregating the $\widehat{IL}_{it}$ and $\widehat{IS}_{it}$ variables in that quarter, respectively:

$$\widehat{IL}_t = \sum_{i=1}^{N(t)} (1 + \sum_{z=1}^{Z} \widehat{\lambda}_z Zone_{it,z})(1 + \sum_{s=1}^{S} \widehat{\beta}_s \; Slope \; Group_{it,s})L_{it}, \tag{16}$$

$$\widehat{IS}_t = \sum_{i=1}^{N(t)} (1 - \widehat{\delta})^{A_{it}}(1 + \sum_{r=1}^{R} \widehat{\eta}_r R_{it,br})S_{it}. \tag{17}$$

To construct the overall house price index in quarter $t$, the estimated constant-quality land price in quarter $t$, namely $\widehat{p_t^L}$, is normalized such that the land price index in the first quarter is 1:

$$\widetilde{p_t^L} = \frac{\widehat{p_t^L}}{\widehat{p_1^L}}. \tag{18}$$

The total constant-quality amount of land $\widehat{IL}_t$ in quarter $t$ is rescaled accordingly, in order to maintain the predicted constant-quality of land values:

$$\widetilde{IL}_t = \widehat{p_1^L} \cdot \widehat{IL}_t. \tag{19}$$

Following this, the constant-quality amount of housing structure prices and the total constant-quality amount of housing structures are normalized and rescaled in a similar way; these are then presented in Equations (20) and (21):

$$\widetilde{p_t^S} = \frac{\gamma_t}{\gamma_1}, \tag{20}$$

$$\widetilde{IS}_t = \gamma_1 \cdot \widehat{IS}_t. \tag{21}$$

The prices and quantities of the aggregated constant-quality of land and housing structures, which were obtained from Equations (18)–(21), are then used to construct the Fisher house price index (Fisher, 1921 [54]). The Fisher index is chosen over the Laspeyres and Paasche indices because the Laspeyres index is positively biased while the Paasche index is negatively biased. A similar procedure is used to form the quality-adjusted land and housing structure indices, and the Fisher chained house price index for the estimated standard builder's model and for the generalized model without land slopes.

The normalized sub-indices for the quality-adjusted housing structures and land that were initially reported in Tables 4 and 5 are now presented in Table 6, together with the imputed aggregate Fisher chained house indices for all three models. The aggregate house price indices from the traditional hedonic model with time-dummy variables and variables from the generalized builder's models that controlled for structural and land characteristics (including land slope) are reported in column 8 of Table 6 for comparison. The estimated traditional hedonic model takes the following form:

$$ln(p_{it}) = \alpha + \sum_{t=2}^{T} \delta_t D_{it} + \gamma_L ln(S_{it}) + \gamma_A ln(A_{it}) + \sum_{z=1}^{Z} \lambda_z Zone_{it,z} + \sum_{s=1}^{S} \beta_s Slope\ Group_{it,s} + \varepsilon_{it} \tag{22}$$

House price indices from the hedonic regression are then constructed as the exponential of $\hat{\delta}_t$. Land and house price indices are also plotted in Figure 3.

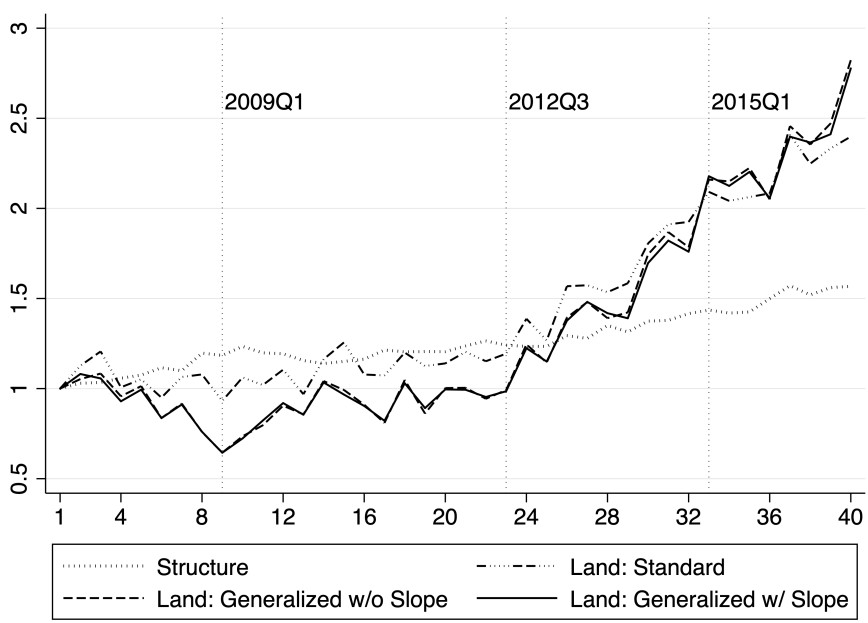

(**a**) Quarterly Constant-Quality Sub-Price Indices: 2007Q1–2016Q4

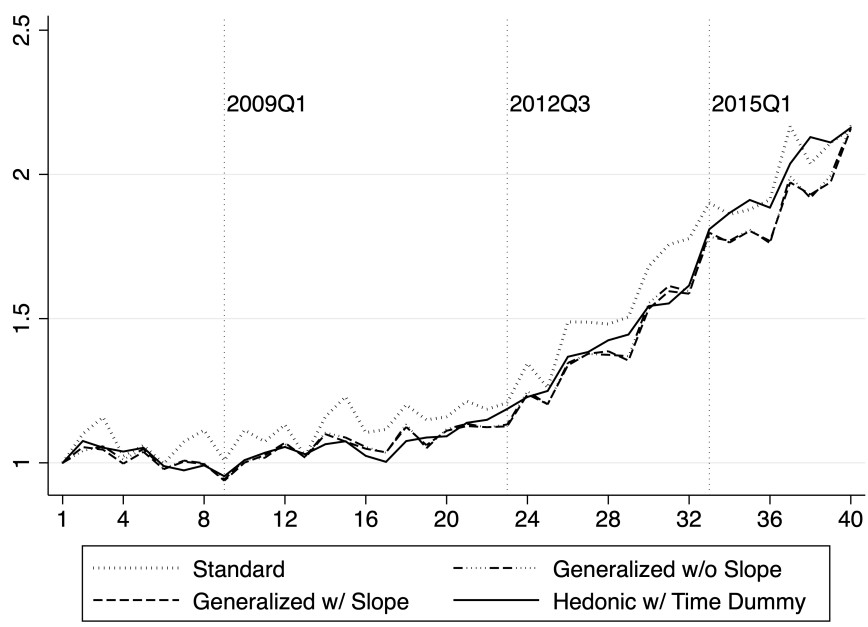

(**b**) Quarterly Aggregate House Price Indices: 2007Q1–2016Q4

**Figure 3.** Constant Quality Sub Price Indices and Aggregate House Price Indices. (**a**) depicts the normalized sub-price indices in Table 6, and (**b**) depicts the normalized aggregate house price indices displayed in Table 6.

It appears that the standard builder's model, namely the one using only the land area, floor area, and decade age, generates higher land price indices up to the fourth quarter of 2014 when compared to the indices of the generalized models. The generalized builder's models show that land prices decreased by about 30% at the end of 2008 compared to the first quarter of 2007, whereas the construction costs of housing structures increased by about 18% in the same time period. This is consistent with the literature stating that the prices of housing structures fluctuate less than the land prices, and that the land and structure values evolve differently over time.

Comparing the land prices from the two generalized models shows that using the land slope as an additional explanatory variable leads to a 4.57 percentage point reduction in the land price index of the last quarter (i.e., equivalent to a 1.62% decrease) when compared to the generalized model without slopes. From what can also be seen in panel (a) of Figure 3, the estimated land prices are almost identical across most quarters, but they do moderately differ for several quarters. Therefore, these results should be interpreted with caution. As the land slope was previously found to decrease the value of houses (Hypothesis 1), the minor changes observed in the land price indices after controlling for land slope imply that the slope composition does not change over time (case 2). Indeed, at the significance level of 0.1, the hypothesis that the land slope mean is the same in the first quarter of 2007 as in the subsequent quarters is only rejected for the fourth quarter of 2009 and the first quarter of 2014; the largest difference in mean slope is 1.26°.

When investigating the aggregate house price indices, we can observe that, in general, the standard builder's model generates larger house price indices (see Table 6 and panel (b) of Figure 3). For the fourth quarter of 2016, the difference between the Fisher chained house price indices of the two generalized models is 1.18 percentage point (i.e., equivalent to a 0.5% decrease after controlling for land slope). Panel (b) of Figure 3 also shows that the house price indices from the traditional time-dummy hedonic regression closely follow the Fisher chained house price indices from the generalized models up to the first quarter of 2015.

## 4. Discussion

The importance of separating the housing structure from the land has been previously well established, but the practical difficulties of separating these two elements remain. Unlike structure, land is not reproducible. Land parcels differ not only in their location and size but also in their slope and other topographical features. Therefore, in order to form reliable constant-quality land price indices, it is necessary to control for those physical attributes of land that can intrinsically limit land use and thus possibly decrease land values.

This paper aimed to demonstrate how a land-specific topographical characteristic—the terrain slope—can be incorporated into the builder's model. Based on a small neighborhood in Auckland where sloped terrain is common, our analysis revealed a so-called slope discount: Having a lower land price per square meter compensates for the difficulty and complexity of building on sloped land. This result should not be directly generalized to other locations with a sloped terrain. Instead, as discussed in Section 2.4, it should be taken into consideration that there are two forces through which the land slope can affect land prices. The land slope may decrease land prices because of the increased complexity and cost necessary to build on such land, but it may also increase land prices due to the potential superior views. Which of these two forces is more influential may depend on the local topography. For instance, in locations where the slope can provide aesthetic advantages, such as a spectacular view of a lake or a mountain, it is reasonable to expect a price premium for sloping sites. Our findings suggest that the land slope has a negligible impact on the quality-adjusted land price index when the composition of the sloped houses sold remains stable over time. Once again, this should be interpreted cautiously, as it does not necessarily imply that the land slope is of no importance to the quality-adjusted land price indices. In 2018, the official magazine of the Registered Master Builders Association

(RMBA) in New Zealand, *Building Today*, reported that, although there continues to be strong demand for flat land, consumers' attention is now turning to sloping land. Please refer to https://www.buildingtoday.co.nz/2018/04/10/the-invisible-costs-of-building-a-house-in-nz/ accessed on 21 March 2019. There is a possibility that, over time, with an increasing amount of new houses being built on sloping land, the effect of including the land slope in quality-adjusted land price indices may become more critical. In addition, for instance, as the subdivision of hilly areas becomes a problem in Los Angeles, there might be fewer houses that are built on sloping sites and sold over time; in such a case, ignoring the variation in land slope composition would lead to biased land price indices.

Our results also seem to support the idea that using the builder's model with only four explanatory variables (i.e., land area, location of the house, floor area, and house age) generates credible overall house price indices and reasonable sub-price indices for the land and the housing structures. However, the moderate change in land price indices after including land slopes may also be a result of our small sample size, since our study area only encompasses three neighboring school enrollment zones in Auckland. It would be also relevant to investigate this effect when applied to a larger spatial context with more sloped observations.

The other limitation of this study and possible area of investigation for future studies is the fact that the model used in this paper has a rather restrictive specification. It assumes that land price differences between school zones and across degrees of land slope do not change over time. However, it is likely that land in the most sought-after school zones may appreciate more than that in other areas. Similarly, the prices of less steep land may increase faster over time than those of steeper land due to the scarcity of such flat land, especially in hillier areas. Therefore, multiplicative interactions between these two variables and time may be important. In addition, sloping land can be subjected to higher risks due to natural hazards. For example, the city of Christchurch experienced extensive soil liquefaction in 2010 and 2011 as a result of a series of large-scale earthquakes. Port Hills, the hilly part of the city, also experienced landslides and rockfalls. Based on this, investigating the interaction between the slope of land and the risk of natural hazards would be an interesting topic for future research.

**Author Contributions:** Conceptualization, Y.H.; formal analysis, Y.H.; writing—original draft preparation, Y.H.; writing—review and editing, G.H.; supervision, G.H. Both authors have read and agreed to the published version of the manuscript.

**Funding:** This research received no external funding.

**Data Availability Statement:** The sales data analyzed during the current study are not publicly available due to the data source's data use policy.

**Acknowledgments:** We gratefully acknowledge the financial support of Jiarong Stella Qian in the acquisition of the data. We are very grateful to Erwin Diewert, Anil Bera, Daniel McMillen, and Sandy Dall'erba for their helpful comments. We would also like to thank Md Shakil Bin Kashem, Haozhi Pan, and Si Chen for ArcGIS support. This paper has also benefited from comments by participants at the Regional Economics Applications Laboratory (REAL) seminars and comments by the anonymous reviewers. The responsibility for any errors is entirely our own.

**Conflicts of Interest:** The authors declare no conflict of interest.

## Appendix A. Slope

A slope represents the rise or fall of the land surface. It is important for a builder to identify whether a land is sloped, as sloped land can be challenging to work on. Mathematically, the slope of a piece of land is expressed as a "rise over run", where the rise is the vertical difference (i.e., difference in height/elevation) between two points on the land area, and the run is the horizontal distance between these two points:

$$\text{slope} = \frac{\text{rise}}{\text{run}} = \frac{\text{vertical difference}}{\text{horizontal distance}}.$$

The percentage rise (%) of a slope is then computed as slope $\times$ 100. The degree (°) of the slope is $\theta$.

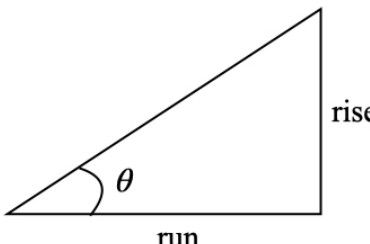

**Figure A1.** Slope.

Table A1 below reports a range of slope classifications used in different countries and different settings.

**Table A1.** Classification of land slopes. (**a**) http://www.fao.org/3/r4082e/r4082e04.htm accessed on 1 March 2019, (**b**) http://sis.agr.gc.ca/cansis/nsdb/slc/v3.2/cmp/slope.html accessed on 1 March 2019, (**c**) https://www.tweed.nsw.gov.au/Download.aspx?Path=~/Documents/Planning/TSC02931_Fact_Sheet_4_Working_with_Sloping_Sites.pdf accessed on 1 March 2019, (**d**) http://www.archcollege.com/archcollege/2016/9/28094.html accessed on 1 March 2019.

| Slope Class | % | ° |
|---|---|---|
| Horizontal | 0–2 | 0–1.15 |
| Very Flat | 2–5 | 1.15–2.86 |
| Flat | 5–10 | 2.86–5.71 |
| Moderate | 10–25 | 5.71–14.03 |
| Steep | >25 | >14.03 |

| **(b) Government of Canada Soil Landscape of Canada** | | |
|---|---|---|
| **Slope Class** | **%** | **°** |
| Little or None | 0–3 | 0–1.72 |
| Gentle | 4–9 | 2.29–5.14 |
| Moderate | 10–15 | 5.71–8.53 |
| Steep | 16–30 | 9.09–16.70 |
| Extremely steep | 31–60 | 17.22–30.96 |
| Excessively steep | >60 | >30.96 |

| **(c) Tweed Shire Council Australia Dwelling Houses** | | |
|---|---|---|
| **Slope Class** | **%** | **°** |
| Flat | 0–10 | 0–5.71 |
| Moderate | 10–21 | 5.71–11.86 |
| Steep | 21–32 | 11.86–17.74 |
| Extremely | >32 | >17.74 |

| **(d) China Dwelling Houses** | | |
|---|---|---|
| **Slope Class** | **%** | **°** |
| Flat | 0–2 | 0–1.15 |
| Gentle | 3–9 | 1.72–5.14 |
| Moderate | 10–24 | 5.71–13.50 |
| Steep | 25–50 | 14.04–26.57 |
| Extremely | 50–100 | 26.57–45 |

Note: This table reports a range of slope classifications used in different countries and different settings. (**a**) reports the slopes commonly used in irrigated fields; (**b**) reports the slope gradients for soil landscapes in Canada; (**c**) reports the slopes associated with building a house in Australia; (**d**) reports the slopes for urban construction suitability in China. The source data for panels (**a**,**b**,**d**) provide the slopes as measured in the percentage rise, whereas the source data for panel (**c**) provides the slopes as measured in degree.

**Table A2.** Estimation results with the alternative land slope classes.

| | Generalized w/ Slope | |
|---|---|---|
| | **Coef.** | **Std.Err.** |
| 2007Q1 | 0.717 *** | (0.054) |
| 2007Q2 | 0.766 *** | (0.051) |
| 2007Q3 | 0.736 *** | (0.068) |
| 2007Q4 | 0.658 *** | (0.051) |
| 2008Q1 | 0.698 *** | (0.052) |
| 2008Q2 | 0.582 *** | (0.043) |
| 2008Q3 | 0.648 *** | (0.090) |
| 2008Q4 | 0.543 *** | (0.060) |
| 2009Q1 | 0.453 *** | (0.055) |
| 2009Q2 | 0.505 *** | (0.048) |
| 2009Q3 | 0.573 *** | (0.044) |
| 2009Q4 | 0.645 *** | (0.045) |
| 2010Q1 | 0.604 *** | (0.041) |
| 2010Q2 | 0.724 *** | (0.059) |
| 2010Q3 | 0.683 *** | (0.064) |
| 2010Q4 | 0.634 *** | (0.059) |
| 2011Q1 | 0.583 *** | (0.055) |
| 2011Q2 | 0.727 *** | (0.065) |
| 2011Q3 | 0.636 *** | (0.061) |
| 2011Q4 | 0.695 *** | (0.068) |
| 2012Q1 | 0.699 *** | (0.049) |
| 2012Q2 | 0.676 *** | (0.043) |
| 2012Q3 | 0.692 *** | (0.046) |
| 2012Q4 | 0.861 *** | (0.043) |
| 2013Q1 | 0.809 *** | (0.045) |
| 2013Q2 | 0.972 *** | (0.052) |
| 2013Q3 | 1.039 *** | (0.060) |
| 2013Q4 | 0.995 *** | (0.061) |
| 2014Q1 | 0.993 *** | (0.062) |
| 2014Q2 | 1.208 *** | (0.081) |
| 2014Q3 | 1.278 *** | (0.082) |
| 2014Q4 | 1.231 *** | (0.057) |
| 2015Q1 | 1.547 *** | (0.080) |
| 2015Q2 | 1.497 *** | (0.071) |
| 2015Q3 | 1.543 *** | (0.070) |
| 2015Q4 | 1.421 *** | (0.063) |
| 2016Q1 | 1.692 *** | (0.095) |
| 2016Q2 | 1.666 *** | (0.080) |
| 2016Q3 | 1.707 *** | (0.085) |
| 2016Q4 | 1.957 *** | (0.144) |
| Decade Discount Rate $\delta$ | 0.066 *** | (0.007) |
| One Tree Hill School Zone | −0.396 *** | (0.014) |
| Double Grammar Zone | 0.534 *** | (0.036) |
| 5 Rooms | 1.043 *** | (0.042) |
| 6 Rooms | 1.061 *** | (0.042) |
| 7 Rooms | 1.274 *** | (0.046) |
| 8+ Rooms | 1.247 *** | (0.047) |
| Flat (0–10%) | 0.036 | (0.026) |
| Steeply Sloped (25–50%) | −0.101 *** | (0.024) |
| Extremely Steeply Sloped (50–70%) | −0.342 *** | (0.055) |
| Adjusted $R^2$ | 0.940 | |
| Log-Likelihood | −40,865.385 | |
| AIC | 81,830.77 | |
| BIC | 82,162.8 | |
| Number of Observations | 5657 | |

Note: This table reports the estimation results for the generalized builder's models using an alternative slope classification. The Selwyn College school zone is the baseline school zone. The reference room category is that of houses with 2–4 rooms. Moderately sloped land (10–25%) is the baseline land slope class. Robust standard errors are reported in parentheses. * $p < 0.10$, ** $p < 0.05$, *** $p < 0.01$.

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
