# Peer review of "More Reliable Land Price Index: Is There a Slope Effect?"

_land, doi:10.3390/land10030261_

Round 1

Reviewer 1 Report

Very well written, and very thorough.

Mino typos on page 9 where the degrees quoted for moderately steep appear to use incorrect numbers: "moderately steep (2-35°), and steep (26-35°)" appear to more probably be: "steep (20-25°), and steep (26-35°)"

Author Response

Following the referee’s comment, the typos on page 9 are corrected.

Reviewer 2 Report

In the scientific literature and studies of valuation practitioners (property appraisers) one can find the results of research on the impact of various physical / technical, legal and neighborhood characteristics of real estate on their market value. In the scientific literature and studies of valuation practitioners (property appraisers) one can find the results of research on the impact of various physical / technical, legal and neighborhood characteristics of real estate on their market value.

For this reason, the topic discussed in the article is interesting. However, in order to find out how much it is worth purchasing such land for a residential house, I suggest using the residual valuation method. It takes into account the prices paid by buyers for finished residential real estate, the costs to be incurred to build a house on a sloping ground and the investor's profit. Such calculations can also be made for flat areas. In my opinion, this will definitely increase the value of the article and allow for a clearer verification of the formulated theses.

The methodology used in the paper is generally correct for price index research, but in my opinion not very convenient for this type of analysis. Hence my recommendation to compare the results obtained also using the residual method of valuation.

Author Response

Good point. The residual method of valuation helps the developers identify the re-development value of a piece of land. It requires an estimate on development costs, including the project's construction cost and investor's profit. As the paper focuses on constructing a more reliable constant-quality land price index, we decide to use the available property transaction data and apply the builder's model. In the builder's model, the use of the exogenous price of the structure per square meter to value the property's structure makes sure the land value is the residual value of the property transaction value.

Reviewer 3 Report

-The title and the analysis are misleading. The authors did focus more on “price index” than on the coefficients related to the “land slope”. What is the main aim of the paper? As most the paper focus/discuss about the price index methods, the reader can hardly see how the main originality/aim of the paper is to focus on the land slope coefficients.

-One of the major problems of the hedonic pricing model approach is not necessarily the multicollinearity, as suggest and discuss in the paper (p.3, lines 90-96), but the omission bias problem. Facing high collinearity between independent variables may be an empirical challenge, but this problem can be fixed, for example, by using principal component analysis (PCA). However, the omission bias problem is a more serious one. Thus, I am wondering why the authors really want to discuss the multicollinearity issue in the paper.

-Why not use the continuous variable for the slope instead of using “categories”? As the results may change depending on how this categorical variable is built (p. 9; lines 278-288).

Minor comments:

-References should be provided for the discussion in the second paragraph of the introduction (p.1; lines 19-25).

-Titles of tables 5 and 6 should be revised. Complementary information should be placed under the table. Same thing for the legend.

-In Appendix A, a “e” is missing in the “slop” equation.

Author Response

Point 1: The title and the analysis are misleading. The authors did focus more on “price index” than on the coefficients related to the “land slope”. What is the main aim of the paper? As most the paper focus/discuss about the price index methods, the reader can hardly see how the main originality/aim of the paper is to focus on the land slope coefficients.

Response 1: Following the reviewer’s comment, the title is changed to “More Reliable Land Price Index: Is There a Slope Effect?”

Point 2: One of the major problems of the hedonic pricing model approach is not necessarily the multicollinearity, as suggest and discuss in the paper (p.3, lines 90-96), but the omission bias problem. Facing high collinearity between independent variables may be an empirical challenge, but this problem can be fixed, for example, by using principal component analysis (PCA). However, the omission bias problem is a more serious one. Thus, I am wondering why the authors really want to discuss the multicollinearity issue in the paper.

Response 2: Multicollinearity becomes more of a concern when the focus is on decomposing the overall house price index into the sub-price indices, i.e., the price index of property structures and the land price index. The high correlation between land area and floor area often leads to volatile and unreasonable sub-price indices estimates. Hence, we follow the empirical evidence and use an exogenous price of the structure per meter squared to value the property's structure to make sure that the land value is the residual value of the property; the estimated land price indices will be more stable and reasonable.

Point 3: Why not use the continuous variable for the slope instead of using “categories”? As the results may change depending on how this categorical variable is built (p. 9; lines 278-288).

Response 3: Categorizing land slope allows the price of land to change nonlinearly with slope. Using a continuous variable for the slope, the slope's marginal effect on price is constant and invariant to the reference slope. With continuous measure, a one-degree increase in slope from 1° to 2° has the same impact on price as a one-degree increase from 90° to 91°. We reckon using slope categories is a better representation of the actual pricing of the slope. We also estimate the builder’s model with an alternative land slope classification. The results presented in Appendix Table A2 (p.22) are consistent with the main results that use land slope classification from the Land Resource Information System (LRIS) New Zealand.

Point 4: References should be provided for the discussion in the second paragraph of the introduction (p.1; lines 19-25).

Response 4: Follow the reviewer’s comment, references are added.

Point 5: Titles of tables 5 and 6 should be revised. Complementary information should be placed under the table. Same thing for the legend.

Response 5: Follow the reviewer’s comment, complementary information for tables is all placed under the tables.

Point 6: In Appendix A, a “e” is missing in the “slop” equation.

Response 6: Follow the reviewer’s comment, the typo in the equation is corrected (p.20).

Round 2

Reviewer 2 Report

I can agree with you but include  the explanation  ( from your responses) to the paper. 

Author Response

Follow the reviewer's suggestion, explanation has been included to the paper on p.2 lines 66-73.